# Exclusion or Inclusion: National Differential Regulations of Migrant Workers’ Employment, Social Protection, and Migrations Policies on Im/Mobilities in East Asia-Examples of South Korea and Taiwan

**DOI:** 10.3390/ijerph192316270

**Published:** 2022-12-05

**Authors:** Yoon Kyung Kwak, Ming Sheng Wang

**Affiliations:** 1Korea Institute for Health and Social Affairs, Sejong-si 030147, Republic of Korea; 2Graduate Institute of Social Work, National Chengchi University, Taipei 116011, Taiwan

**Keywords:** blue-collar workers, labor migration, employment policies, precarity, public health

## Abstract

Low fertility rates and an aging society, growing long-term care needs, and workforce shortages in professional, industrial, and care sectors are emerging issues in South Korea and Taiwan. Both governments have pursued economic/industrial growth as productive welfare capitalism and enacted preferred selective migration policies to recruit white-collar migrant workers (MWs) as mobile elites, but they have also adopted regulations and limitations on blue-collar MWs through unfree labor relations, precarious employment, and temporary legal status to provide supplemental labor. In order to demonstrate how multiple policy regulations from a national level affect MWs’ precarity of labor in their receiving countries, which in turn affect MWs’ im/mobilities, this article presents the growing trends of transnational MWs, regardless of them being high- or low-skilled MWs, and it evaluates four dimensions of labor migration policies—MWs’ working and employment conditions, social protection, union rights and political participation, and access to permanent residency in both countries. We found that the rights and working conditions of low-skilled MWs in Korea and Taiwan are improving slowly, but still lag behind those of high-skilled MWs which also affects their public health and well-being. The significant difference identified here is that MWs in Taiwan can organize labor unions, which is strictly prohibited in Korea; pension protection also differs between the nations. Additionally, an application for permanent residency is easier for high-skilled migrant workers compared with low-skilled MWs and both the Korean and Taiwanese immigration policies differentiate the entry and resident status for low-skilled and professional MWs from dissimilar class backgrounds. Policy recommendations for both countries are also discussed.

## 1. Introduction

Exploring the complex and dynamic phenomena of migration, the classic review and appraisal of the theories of international migration from Massey et al. (1993) elaborated neoclassical economics (e.g., micro vs. macro), new economics, a dual labor market, network theory, institutional theory, cumulative causation, and migration systems theory [1], to an interdisciplinary interpretation of the migration and im/mobilities perspectives on transnational migration [2,3]. Regardless of being from a micro-, meso-, eso- or macro-level, or from an exploration of the sending and receiving countries’ experiences, their structural factors/intersectionalized regimes or policies, social determinants affecting individual factors for international migrations and MWs’ health, wellbeing, and futures cannot be neglected. Scholars have argued that it is crucial to reveal the power geometries of migration, care labor, and precarity chains to understand the power relations in transnational flows and movement, as well as the im/mobilities of the globalization process, particularly regarding political sovereignty at the state-led level [1,2,3,4,5,6,7,8,9]. In other words, (1) the push factors (such as a poor living environment and low employment opportunities) from the sending countries, (2) the pull factors (such as a high salary) from the receiving countries, (3) the socio-economic/political structure which connects the sending and host countries, and (4) the people’s willingness and motivation which react to the structural factors from migrating, are the four core elements for international migration [1,8]; however, the stepwise or the cycle or precarity of migration in and between origin and host countries, as well as the aspirations of the actors who respond to these same forces by staying, running away, deregulating/regulating, or experiencing deportation must also be included [3,5,8].

Our research conceptual model was synthesized from prior migration research [1,2,3,4,5,6,7,8], as the bottom of Figure 1 shows, where in the initial stage of the precarity of migration, the limited choice of a transnational migrant destination is affected by multi-level factors, including MWs as individuals, households, networks, local agents, and the socio-economic/political development of the sending countries and the national regulations in receiving countries [4,5]. After arriving in the receiving countries, the precarity of labor (such as the working conditions and salaries) is again closely intertwined with the MWs’ individual factors and is affected by intersectionalized policies in the receiving countries, which also in turn have impacts on MWs’ current equality of health and wellbeing as well as the precarity of their futures [5]. Therefore, differing from research that places more emphasis on MWs’ individual desires and capacities for mobilities in sending countries, this article focuses more on how the national differential regulations in receiving countries have impacts on the inclusion of high-skilled and the exclusion of low-skilled MWs separately, that in turn affect MWs im/mobilities as the Figure 1 right red cycle shows [6,7,8].

In addition, the regulation/deregulation of migrant workers’ employment and migration policies, as well as the segregation of high-skilled/white-collar/professional and low-skilled/blue-collar MWs, have been closely intersectionalized with the economic development of sending countries and receiving countries and with the political sovereignty and ideology behind recruiting so-called “talents” and “supplemental labor” [9,10,11]. Global migration, with its continuous spatial movement, is significantly related to economic insecurity and socio-economic class im/mobilities, particularly for those low-skilled MWs in sending countries [4,5,12,13]. The importation, regulation, and segregation of MWs reflect the receiving governments’ emphases and concerns about national economic growth and the competitiveness of domestic industries rather than the working conditions and social rights of MWs [14,15,16,17]. Therefore, nations place more regulations on MWs’ labor/employment and restrictions on their migration/citizenship. This is particularly the case in the East Asia context which is classified as a developmental/productivist, welfare capitalism/regime and which emphasizes industrial/economic growth as a prior goal [17,18].

South Korea (hereafter Korea) and Taiwan were two of the Four Little Dragons, which experienced rapid industrialization and continued high growth rates between the early 1960s and 1990s. In the process of transforming from traditional industries to high-tech industries and service sectors, both the Korean and Taiwanese societies not only developed their economies but also changed the structures of their labor markets. One of the significant changes to their labor markets has been the growth of foreign-born migrant workers (MWs), especially low-skilled migrant workers in both countries. Specifically, domestic labor wages have increased, and locals have started to become reluctant to work in 3D (i.e., dirty, difficult, and dangerous) labor for lower payments, such as in major construction projects [19,20,21]. That is why the construction and manufacturing industries have had no choice but to look for alternatives to fill the shortages of labor, and accordingly, the demand for low-skilled MWs has increased substantially. This is how both countries have transformed from being labor-sending countries to labor-receiving countries. In recent years, faced with the rise in high-tech industries and the service sector, both the Korean and Taiwanese governments have also attracted high-skilled MWs, to increase their national competitiveness and cope with a knowledge-based society in the coming years [9,22].

## 2. Methods

International comparative data on migrant workers are not standardized or available, and most of the data presented in this article derives from various national sources. Information from government annual statistics, secondary literature, policies analysis and international comparative studies was combined. Particularly, we analyze policies related to employment differential regulation, laws on social protections, restrictions on union rights and political participation, and immigration control. We attempt to examine the MWs’ employment and migration policies in Korea and Taiwan and to analyze how different forms of national sovereignty, regulations on employment, social protections, union rights and political participation, and migration policies affect the growth of high-skilled and low-skilled MW populations as well as the equality of their health and wellbeing. We begin by (1) exploring the growing trend of MWs to observe different growth rates of high-skilled/white-collar and low-skilled/blue-collar MWs in Korea and Taiwan; (2) comparing their employment policies to see the similarities and differences between high-skilled and low-skilled MWs based on the following dimensions: the working conditions, freedom of choice in employment and social protection; (3) examining union rights and political participation; and finally (4) distinguishing the levels of acceptance of permanent residency and family reunification in the immigration policies of both countries.

This article contributes by highlighting how the national regulations of employment, social protection, union rights and political participation, and immigration policies impact the im/mobilities and the rights of low-skilled and high-skilled MWs under the competition of globalization. We also point out how low-skilled MWs’ social rights are slowly improving due to the advocacy of vulnerable local unions of MWs, the United Nation’s (UN) influence, and the competitive demands of MWs from other developed countries.

## 3. Results

South Korea has adopted a selected approach towards migrants in the previous decades. In the 1990s, the Korean government began implementing policies for low-skilled migrant workers, the so-called Industrial Trainee Program (ITP), which was replaced with the Employment Permit System (EPS) in 2004, in order to fill the shortage of labor in 3D industries (i.e., dirty, difficult and dangerous). The recruitment occurred through bilateral agreements with 16 Asian developing countries such as Bangladesh, Myanmar, Nepal, etc. [23]. The number of high-skilled workers is not large, and foreign-language instructors account for the largest share [23]. Unlike the low-skilled workers, the high-skilled ones have more autonomy in terms of changing their workplaces, family reunification and so on. Faced with a low birth rate and aging society, however, a new initiative has been introduced that paves the way for implementing a new immigration policy in 2018 [24]. In this case, the supply of high-skilled work is strongly encouraged, because it is believed that such workers can contribute to increasing the national competitiveness and coping with the knowledge-based society of the coming years [22]. At the same time, they have lifted restrictions on the duration of their stay and the number of workplace changes, etc. (OECD, 2019).

For Taiwan, there were several waves of migration from Mainland China and most immigrants were Han people, although they speak different local languages. Since 1990, due to the strict immigration policy, most immigrants coming to Taiwan have been female brides from Mainland China, Vietnam, the Philippines, and Indonesia through marriage or as dependent relatives during the last two decades. As of 5 November 2016, the period of stay for blue-collar MWs was three years. After which, these workers were required to leave Taiwan for at least one day. The working periods of MWs were not calculated in the consideration of their applications for permanent residency. The migration policy in Taiwan is friendlier to high-skilled/professional MWs than to low-skilled MWs because the stays of the working periods for low-skilled MWs are not considered to qualify for permanent residency. Blue-collar migrant workers can only obtain temporary residency. Similar to many developed countries, Korea and Taiwan’s differential employment policies for migrant workers have also moved forward to “class selection” and a global “convergence” of immigration policies which take the economic return of potential migrant workers into consideration [9]. Below is our analysis in detail through a comparative perspective.

### 3.1. Trends of Migrant Workers in Korea and Taiwan

The compositions of MWs, especially foreign-born, low-skilled workers, vary. In Korea, we found two groups of low-skilled workers, including ethnic Koreans with foreign citizenship who lived in China or the states of the former Soviet Union and migrant workers mainly from developing southeast Asian countries arriving either through the ITP or EPS [19,23]. To be specific, the government has taken the stance that ethnic Koreans with foreign citizenship can blend in with Korean workers more easily than foreign workers due to similarities in their appearances and shared Korean culture and norms [19,25]. That is why the government has strongly encouraged ethnic Koreans with foreign citizenship, especially those with Chinese citizenship, called “chosunjok”, to enter and work in Korea. It was not until 2007 that the Korean government took further action by revising the Overseas Koreans Law and implementing the Work Visit visa system (H-2) for overseas Koreans from China and the former Soviet Union states [19]. Therefore, ethnic Koreans, including the “kyoryoin”, the lineal descendants of ethnic Koreans who were forcibly deported to Central Asia due to the Soviet Policy in 1937, were finally eligible for the H-2 visa, a special work and residence permit [25,26,27,28]. Overall, these low-skilled migrant workers can only work in designated industries such as agriculture, manufacturing, construction, and the service industry [19].

Additionally, the Korean government introduced the ITP in 1993. Under the ITP, low-skilled workers, mostly male workers, from developing countries arrived in Korea to work in the 3D industries for three years and later were obliged to return to their home countries at the end of their contracts [29]. As they were employed as trainees rather than workers, their legal status as workers made them very vulnerable [30]. To address these problems, the ITP was replaced with the EPS (Employment Permit System) in 2004 [28,29]. Under the EPS, low-skilled workers can be re-employed in Korea after 6 months from the day of departure, but in this case, the employer must request a visa extension for up to one year and ten months from the Ministry of Justice before the workers whose visas are about to expire return to their home country at the end of their contract [31].

In the context of Taiwan, it was not until 1989 that the government allowed low-skilled migrant workers who were permitted to enter and work under three principles: (1) the MWs were treated as supplemental laborers, (2) a quota of MWs was enforced, and (3) work-permits were limited to specific industries, such as the major construction and manufacturing industries, to avoid impacts on the local employment of domestic laborers [32,33]. The regulation and application procedures for low-skilled foreign workers were more restricted, complicated, and time-consuming, compared to those for white-collar/high-skilled foreign workers. As of 2017, there were a total of 720,000 Taiwanese working overseas and 31,000 foreign professionals working in Taiwan [34]. Meanwhile, low-skilled MWs were still only allowed to work in the originally permitted industries such as manufacturing, construction, and the care industry. Only if the low-skilled MWs were proved to be victims of human trafficking could they then shift to any kind of industry under the special permission of the Ministry of Labor in Taiwan. From Table 1, we can find that the number of high-skilled/professional MWs in Korea was triple, and the low-skilled MWs in the industrial sector was double from 2002 to 2017. The number of high-skilled/professional MWs in Taiwan has been growing slowly but the low-skilled MWs are growing fast, regardless of being in the industrial or in the welfare sectors. In sum, Korea recruits more professional MWs and Taiwan recruits more low-skilled MWs.

### 3.2. Employment Policies

#### 3.2.1. Working Conditions

##### Korea

In the context of Korea, there are similarities between high-skilled and low-skilled MWs’ experiences, which both include long working hours. Specifically, the average number of working hours for low-skilled workers was 58.7 h per week in 2013, whereas the average for Koreans was 43.1 h per week in the same year [35,36]. Although no specific studies have investigated the working hours of high-skilled MWs, a book published by Eric Surdej, a former president of LG Electronics France, showed that long working hours are very common for such workers [37].

When it comes to differences between the two groups, the low-skilled workers have higher rates of adverse working conditions compared with the high-skilled ones, though almost no studies have examined the working conditions of high-skilled MWs. The main issue is the significant wage gap between native and foreign-born workers, which reached up to 55 percent, the highest among the Organization of Economic Cooperation and Development countries [38]. It was reported that the average monthly income of foreign-born workers was USD 1380 (KRW 1.55 million) in 2013 [35], and that many experienced delays in the payment of their wages [27]. The reasons for such discriminatory treatment include the negative perceptions of the low-skilled workers’ countries of origin—Asian developing countries, and the total disregard for physical labor in Korea [27,39] In addition, MWs tend to experience subtle discrimination from their ethnic Korean colleagues because of prejudice and negative perceptions embedded in Korean society [30,40]. In terms of living conditions, one study showed that 44.2 percent of low-skilled workers lived in dormitories [27]. Dormitory living is the preferred option because the relatively lower cost helps the residents to save money and send remittances to family members in their countries of origin. From the perspective of employers, it is easier to control and manage low-skilled workers living in dormitories. The problem is that the workers often live in rooms of less than 6.6 m^2^ and share them with at least three roommates [19].

##### Taiwan

The high-skilled MWs have higher salaries and the freedom to choose their accommodations compared with low-skilled MWs. Foreign white-collar workers are treated as special professionals or technical workers in Taiwan and there are seven categories of work, including specialized or technical work, full-time teachers teaching courses in foreign languages in short-term classes registered for supplementary schooling, school teachers, directors of businesses invested in or set up by overseas Chinese or foreigner(s), religious, artistic, and show business workers, and sports coaches and athletes. Compared to low-skilled MWs, the restrictions and regulations on hiring professional MWs are relatively loose; therefore, some employers are importing low-skilled workers but apply for professional white-collar visas for them [32]. Statistics show that “full-time teachers teaching courses in foreign languages in short-term classes registered for supplementary schooling” and “specialized or technical work” were the two categories of work that employers applied for false white-collar worker visas for workers performing blue-collar labor [32].

In contrast, the low-skilled MWs earn lower salaries. Low-skilled MWs in productive sectors and welfare institutions (such as nursing homes) are protected by the minimum standards under the protection from the Labor Standard Act in the employment codes, for which the monthly income was raised from around USD 733 (TWD 22,000) in 2018 to USD 770 (TWD 23,100) in 2019, and USD 790 (TWD 23,700) in 2020. Only live-in migrant care workers are excluded from the Labor Standard Act and do not qualify for the minimum wage. The monthly income for these workers was around USD 567 (TWD 17,000) in 2019 and this salary has not been raised since 2015.

According to the Foreign Labor Management and Application Survey, a low-skilled MW’s total average monthly income is around USD 926 (TWD 27,788) and their monthly total average working hours are 195.3 h (daily working hours of near to 9.76 h) in the productive sector. In addition, the total average monthly income is around USD 664 (TWD 19,927) for live-in care workers and the daily working hours are 10.2 h [41]; however, these live-in care workers do not receive any additional compensation for overtime work because of the lack of a clear boundary between care work and personal rest time in the private sector/households.

The employers of most low-skilled MWs in productive sectors and welfare institutions provide dormitories for these MWs and the fee is deducted from their monthly salary. Only migrant care workers living in private households do not need to make this kind of payment. Many low-skilled MWs are provided with accommodation and food by their employers, but the costs are deducted from their monthly salary; however, the living space is typically under 3.2 m^2^ and is often inside factories without a clear separation between work and rest territories. Several serious fires occurred in factories causing MWs’ deaths due to a lack of adequate separation between the factories and their dormitories [20]. Later, the Ministry of Labor passed the law to enforce the separation of working areas in the factory and living areas in the dormitory. From analyzing the 1955 hotline for low-skilled MWs related to labor consulting and disputes, the first three factors separately were the maltreatment from employers, illegal work outside the work permit, and disputes over housing and food [20,41]).

#### 3.2.2. Freedom of Choice and Employment Protection

##### Korea

All foreign-born workers in Korea are protected by domestic labor laws and, thus, have basic rights such as a minimum wage and other labor rights, but low-skilled workers entering through the EPS face disadvantages when changing workplaces [27]. Originally, changes of workplaces were strictly constrained, especially through the EPS during their contract period, and such restrictive measures led to the unexpected outcome that the number of irregular migrant workers increased. For example, the workers needed to repay large fees to recruitment agencies in their country of origin [19]. In this regard, if the monthly income was not sufficient to repay the large fee to these agencies, the workers had no choice but to search for a new job with a higher salary [19]. With a consideration for such problems, the government relaxed the rules in 2009, allowing low-skilled workers to change their places of work three times within three years, with the permission of the employers [24]. Of course, if an employer has a reason, then the workers are eligible to change workplaces; however, the biggest problem is that the workers themselves have to prove their reasons, such as sexual violence, battering, rape, exploitation, a delay in the payment of wages, an abuse of human rights, etc. Critics argue that the current system produces irregular migrants. Specifically, they may be subject to control if they change workplaces more than three times and leave workplaces due to repeated violations of human rights; thus, migrant workers in Korea are at risk of becoming irregular workers due to the strict restrictions on their workplace movements.

##### Taiwan

High-skilled MWs are recruited directly by their employers. Moreover, compared to low-skilled MWs, white-collar/professional MWs have more power to negotiate their working conditions and salaries. Most blue-collar/low-skilled MWs are recruited through brokers in their home country and the MWs need to pay a one-time brokerage fee to the broker in their home country and a monthly service fee to the local agent in Taiwan. According to Article 53 of the Taiwan Employment Service Act, low-skilled MWs are not allowed to shift to new employers or new workplaces. Only when certain circumstances have arisen or exist, such as when an employer is unable to pay or dies, the company closes, the migrant workers are abused (e.g., physical, verbal or sexual abuse), in cases of labor disputes or illegal hiring, and so on, can a foreign worker work shift to a new employer or engage in new work upon the authorization of the central competent authority [38].

Article 59 in the Taiwan Employment Service Act lists the following regulations: (1) the original employer or the one who was intended to be taken care of by the employed foreign worker has deceased or emigrated; (2) the vessel a MW works on has been seized, has sunk, or has been under repair to compel a discontinuation of the work; (3) a discontinuation of work is caused when the MW’s original employer has closed a factory, suspended the business, or failed to pay the wage/salary under the employment contract resulting in the termination thereof; or (4) other than the above, similar circumstances not attributable to the employed MW. The current regulation on low-skilled MWs shows that the Taiwan government treats MWs as supplemental labor, but gives employers more power to stabilize the employment market and avoid MWs changing jobs arbitrarily. The central competent authority promulgates the procedures governing shifts to new employers or new work [36]. In addition, the agents are also eager to engage in new job transfers due to the one-time transfer fee charged to both the private employer and migrant workers, particularly due to the severe shortage of the care workforce.

### 3.3. Social Protection (Labor Insurance, Health Rights, and Pension)

#### 3.3.1. Korea

The subscription status for migrant workers varies depending on the visa status and type of visa. There are four forms of insurance: industrial accident compensation insurance, national health insurance, national pension, and employment insurance [22,31]. First, all migrant workers are entitled to industrial accident compensation insurance. This insurance allows employees to claim benefits in the case of an accident, disease, etc., during their work [22,27], although irregular migrant workers can benefit from the insurance, in reality, it is these workers who are less likely to report an industrial accident, because such a report could reveal their presence in Korea and eventually endanger their stay [22].

Second, national health insurance is one of the major social insurance programs that pays the medical costs of the insured [30]. If workplaces join and cover the insurance fee, then workers can join as insured employees; however, the workplaces for small business employees, private caregivers, and domestic helpers do not cover the industrial accident compensation insurance. These workers can join for self-employed insurance and can apply without their employer’s consent if they suffer from work-related injuries; however, if migrant workers have other health insurance to cover their health benefits, then they can unsubscribe from national health insurance [22].

Third, the national pension benefits are available to foreign workers on the principle of reciprocity [27]. The nationals from China, Philippines, Uzbekistan, Kyrgyzstan, Mongolia, Sri Lanka, Indonesia, and Thailand, where Korea has made social welfare agreements, are required to subscribe to the employee or self-employed national pension, but those from Kyrgyzstan, Mongolia, Sri Lanka, Indonesia, and Thailand are not eligible for the self-employed national pension. Those from other countries, such as Vietnam, Cambodia, Pakistan, Bangladesh, Nepal, Myanmar, and East Timor cannot subscribe to the national pension. Fourth, employment insurance is voluntary, and not mandatory, for all foreign workers.

Interestingly, there are unique insurance programs only designed for low-skilled foreign workers through the Employment Permit System (EPS), such as the Departure Guarantee Insurance (DGI), Wage Payment Guarantee Insurance (WPGI), and Return Cost Insurance (RCI). Under the DGI, a payment can be claimed when a MW returns to their home country at the end of their contract or changes their workplace after one year of work in Korea. The WPGI allows low-skilled migrant workers to file administrative litigation for different wages with Seoul Guarantee Insurance. When they do so, they can receive unpaid wages, up to a maximum of USD 1780 (KRW two million), as stated in the Standards Labor Contract. The RCI, purchased by employees themselves, covers the costs of an employee returning to his or her home country. The insurance fee varies depending on the country of origin, the airfare cost, and other relevant costs. A worker can claim the payment upon the expiration of their labor contract at the time of departure.

#### 3.3.2. Taiwan

Local laborers receive two-tier social security from the first-tier Labor Insurance Act as social insurance and the second-tier new labor pension plan as an individual account. White-collar professional workers and blue-collar migrant workers are protected under the Labor Standard Act. Only live-in care workers do not receive protection; however, only MWs who obtain permanent residence and qualify for the Labor Standard Act are entitled to these two-tier pension schemes. Otherwise, those MWs who obtain short-term work permits are only entitled to Labor Insurance and Employment Insurance and are entitled to claim benefits in the cases of accidents or injuries during their work; however, they are excluded from the second-tier new labor pension plan. Furthermore, live-in care workers are the most vulnerable population who are excluded from both Labor Insurance and Employment Insurance. Employment insurance is mandatory for low-skilled MWs in productive sectors and nursing homes but is not applicable for live-in care workers.

In addition, all MWs are covered by the National Health Insurance (NHI). As long as MWs stay with a regular employer after being issued an alien residence permit, the employers have those MWs enroll in the NHI. Moreover, only high-skilled MWs can bring their family members with them and family members who have established a registered domicile in Taiwan for a waiting period of at least six months can participate in the NHI.

However, while low-skilled MWs are entitled to National Health Insurance, due to the shortage of translators in health clinics, language and communication barriers, special working types, imbalances in the power relationships between employers and employees, and a fear of losing a job due to sickness during a contract, they have difficulties with utilizing the NHI service, even the though medical services in Taiwan are better than those in their home countries. In certain cases, some MWs diagnosed with cancer or suffering from occupational injuries who could not continue to work have been deported by their employers or agents for other reasons before medical treatment and recovery. Nevertheless, low-skilled MWs are often stereotyped by the media as being hosts of infectious diseases, being unhygienic, and lacking in health knowledge and activity. Among the low-skilled MWs, fishermen are the most vulnerable and require access to health services due to their working conditions [42,43].

### 3.4. Union Rights and Political Participation

#### 3.4.1. Korea

When it comes to the right to vote, it was not until 2006 that those aged 19 and over with permanent residency in Korea were given the right to vote in local elections, based on the amendment to the Public Official Election Act in 2005 [22]. Through this, Korea became the first Asian country to allow foreign-born people with permanent residency to participate in local elections. According to Article 10 of the Departures and Arrivals Control Act, they can register with the Alien Registration Records of their local authorities three years after obtaining their permanent residency [22]. However, critics argue that a considerable number of eligible foreign-born residents with permanent residency are not aware of their eligibility in local elections [44]. This is because the pamphlets sent to foreign-born voters are written either in English or Chinese; therefore, migrants who speak other languages may not be able to fully understand their meaning [45]. Additionally, promotions to foreign-born voters are often carried out by the local Multicultural Family Support Centre, meaning that those who do not use the centers and do not belong to a multicultural family may have difficulty obtaining information from other sources [44].

The Korean government tries to foster an environment where foreign-born naturalized Koreans can raise their voices, increase their political participation and even run for election. For example, one migrant woman, a Mongolian-born naturalized Korean, was elected as a proportional representative of the local assembly of Gyeonggi for the Grand National Party in 2010 for the first time in history [22]. Two years later, in 2012, one migrant woman, named Lee Jasmine, was also elected as a proportional representative of the National Assembly for the ruling party [46].

Such a growing diversity of the population in the realms of politics does not necessarily mean that those with permanent residency enjoy the same political participation as Korean citizens. Specifically, foreigners are not allowed to participate in any forms of political activity other than those specifically permitted by the Access to Immigration Act [47]. That is why foreign students in Korea who are funded by the Korean government are required to sign a pledge that they will not participate in political activities such as establishing or joining political parties, participating in political demonstrations, publishing political articles, or making political declarations [46]. With regard to such general prohibitions on political activities, Kim (2008) criticizes the lack of clarity over the meaning of ‘political activities,’ arguing that the Act infringes on the basic rights of foreigners, such as a freedom of expression, freedom of speech and publication, and freedom of assembly and association [47] (p. 5). Being aware of such a prohibitive environment may have led migrants with spousal visas or permanent residency to feel inhibited about expressing their political opinions and to avoid voicing interest in changing discriminatory policies or laws.

Finally, it was not until 2015 that the High Court in Korea ruled that migrant workers could organize labor unions. In fact, in 2005, one hundred migrant workers launched a labor union, but the Ministry of Employment and Labor argued that irregular workers were not qualified as members of the union. Thus, for the last decade there has been a struggle to secure the rights of migrant workers both inside and outside the courts. In other words, it took ten years to legalize the labor unions in Korea. Their history of resistance against the Ministry of Employment and Labor has contributed to increasing the depth and breadth of labor rights in Korean society. In 2018, three years after the establishment of the union, the number of members increased from 100 in 2015 to 500 in 2018, not including those who joined the union but had since left the country. The problem is that the decision about whether workers can stay for up to one year and ten months longer is up to the employers’ demand; therefore, workers have no choice but to wait to see if their employers’ consent to their joining the union or not.

#### 3.4.2. Taiwan

The Taiwan Labor Union Act never prohibited migrant workers from participating in local labor unions; however, the local Taiwan labor unions (LTLUs) have frequently feared that migrant workers may take local laborers’ work opportunities away and have retained hostile attitudes. Therefore, no MWs have attended any of the LTLUs and the LTLUs seldom fight for the MWs’ working conditions and their rights. As of 2010, when the Taiwan Labor Union Act was modified, MWs were allowed to organize unions to represent their own working rights. Two MWs’ labor unions were organized by the most vulnerable categories of MWs, including marine fishing/netting workers and household assistants and nursing workers. In the beginning, the first migrant workers’ union, called the Yilan Migrant Fishermen Union (YMFU), was initiated and organized on 25 May 2013 by a Taiwanese woman, Allison Lee. Miss Lee had been involved long-term in the disputes of Filipino fishermen fighting for a minimum wage, annual leave, overtime payments, and health rights. In 2018, there were 100 Indonesian members and 20 Filipino members of the YMFU [41,42,43].

The second migrant workers’ union, the Domestic Caretakers Union Taoyuan (DCUT), was organized by migrant care workers from the Philippines in 2017. Although the members of these two unions were made up of the most vulnerable populations among the MWs, they have organized together to advocate and protect their own working and health rights and to allow local people and MWs to hear their voices.

Until now, foreigners who were visiting or residing in Taiwan were not allowed to participate or engage in any forms of political activities or employment that was different from the purposes of their visits or residence until they attained permanent residency, according to the Immigration Act; however, after amending the Immigration Act in 2007, acts of filing petitions or imitating a lawful assembly and procession by those aliens who resided legally were no longer subject to the foresaid restriction. Therefore, MWs do not need to be concerned about deportation by the Taiwan government because of taking to the street to fight and advocate for their own working and social rights [48].

### 3.5. Migration Policies

#### 3.5.1. Permanent Residence

##### Korea

In Korea, the type of visa determines the level of access to permanent residency. In other words, high-skilled workers are more likely to achieve Korean permanent residency than low-skilled ones. The high-skilled workers include foreign investors who have invested more than $500,000 and those who employ more than five Korean nationals [27]. Since 2007, the government has lowered the requirement for professional workers to attain permanent residency, so that those with special skills in science and management and an income three times the GNI per capita are eligible for receiving permanent residency, regardless of the length of their stay in Korea [22]. Additionally, ethnic Koreans with H-2 visas can apply for permanent residency after working in certain sectors such as manufacturing, agriculture, or the livestock industry for more than four years and earning an income equivalent to the GNI per capita.

In contrast, the opportunities to achieve permanent residency are very limited for low-skilled workers through the EPS, although new a measure was introduced in 2012. To qualify for this measure, workers must work in one workplace until their first work permit expires and then should be allowed to re-enter Korea three months after their departure from Korea [27]. To qualify, workers should be employed in the agriculture and fishery industries or small manufacturing companies [31]. If so, they are then entitled to obtain permanent residency. It is estimated that around 10,000 low-skilled workers have benefitted from the recently introduced measure [39].

##### Taiwan

In 1992, blue-collar MWs were only allowed to stay for two years and could extend for one year, making their total stay three years (see Table 2). Later, in 1997, Taiwan’s government changed the basic stay from two to three years and allowed an extra extension of two years, increasing the total stay from three years to five years. In 2002, the total stay period was extended to six years but required a departure from Taiwan for at least 40 days. To solve the shortage of labor, Taiwan’s government allowed two extensions and a maximum stay of nine years in 2007 and three extensions with a maximum stay of twelve years in 2012 for all foreign workers. The government also shortened the required length of departure from at least 40 days to 1 day, and eventually canceled the requirement to leave Taiwan for one day in November 2016. As of 5 November 2016, the period of stay for blue-collar MWs was three years. After which, these workers were required to leave Taiwan for at least one day. The working periods of MWs were not calculated in consideration of their application for permanent residency [45].

Furthermore, facing a shrinking workforce of caregivers and increasing costs of caregiving, Taiwan’s government extended the maximum stay for live-in migrant care workers from 12 to 14 years in October of 2015. According to the new rules, low-income families no longer needed to pay employment security fees if they chose to hire live-in migrant care workers to care for the elderly. However, the maximum stay for MWs in the industrial sector is still twelve years.

Under pressure from an outflow of skilled talent from Taiwan and the challenges of recruiting international professional talents to Taiwan, Taiwan’s government considered easing up the relatively strict immigration policy and creating an “immigrant-friendly” environment to attract more professional talent and mid-level skilled workers in 2017. Consequently, Taiwan announced the “Act for the Recruitment and Employment of Foreign Professionals” on November 22nd in 2017 and placed the Act into practice in February of 2018, to address the workforce shortage issue and boost national competitiveness.

In the past, foreign art workers and foreign teachers hired for short-term tutoring who were engaged in professional work in Taiwan were required to apply for a work permit through a written application. Since the mid-term of October in 2018, the Ministry of Labor opened a platform for 24 h online applications to save time. The Ministry also shortened the processing period to within seven working days and the evaluation process to five working days. Employers and foreign art workers themselves could then apply for working permission from the Ministry of Labor. Furthermore, the biggest difference now is that the license period can be extended by up to three years to five years when hiring special foreign professionals for professional work.

The Executive Yuan passed Taiwan’s New Economic Immigration Bill (NEIB) on November 29 in 2018 to recruit foreign professionals and mid-level foreign technicians to strengthen the workforce and national development without affecting the wage levels or taking local job opportunities away [46]. Only MWs in the productive sector whose monthly salaries reach USD 1380 (TWD 41,393) and those in the welfare sector whose monthly salaries reach USD 1067 (TWD 32,000) are qualified for the NEIB, based on the average monthly salaries of local Taiwanese laborers. If the monthly salary reaches the criteria mentioned above, low-skilled MWs achieve the status of “mid-level technicians” and can apply for permanent residency under certain circumstances.

However, the total average monthly income of MWs in the productive sector was around USD 926 (TWD 27,788) and USD 664 (TWD 19,927) for live-in care workers in 2017 [34,35]. MWs must work in Taiwan for more than 6 years (at least 183 days per year), have a high school degree or above, and have an age of 40 or younger. Moreover, MWs must have relevant professional licenses and their salaries should exceed the basic threshold. In addition, the scope of openness is limited to “technicians and assistant professionals, skilled and mechanical equipment operators, or technical manpower identified by the competent authority”. In the future, industry quotas and control of the total amount will be established. Not all types of work are accepted. Due to the limitation on MWs shifting to new jobs, employers are always reluctant to actively increase the monthly salaries of low-skilled MWs. Without amending Article 53 in the Employment Service Act to allow MWs to compete in the local labor market freely, an application for permanent residency through a transition from a low-skill to mid-level skill status is still difficult and often even impossible for MWs.

#### 3.5.2. Family Reunification

In Korea, there is no restriction for bringing and/or accompanying family members into Korea for professional workers and those with permanent residency and their families; however, low-skilled workers are not permitted to accompany their family members [18,21]).

In Taiwan, family reunification is not allowed for blue-collar/low-skilled MWs but is permitted for white-collar MWs’ family members. Low-skilled MWs are also not allowed to be married to a Taiwanese unless they either stop their contract or finish the contract and return to their home country to start the marriage procedure [43].

## 4. Discussion

Like other developed countries, foreign professionals and immigrant investors are the prioritized and targeted population for recruiting and advocating for permanent residency and legal citizenship applications. In their immigration policies, both Korea and Taiwan differentiate between the entry, residence, immigration statuses, and social and political rights of low-skilled and high-skilled MWs based on their skilled/professional level and substitutability. In addition, their immigration policies for high-skilled MWs have changed and become friendlier to attract talent and outperform other countries in winning global “brains” such as IT specialists. In addition, permanent residency is easier for white-collar/professional MWs to apply for; these workers are treated as “elites” and “talent”, compared to blue-collar/low-skilled MWs, who are treated as “commodities” and “disposable labor”. Although Taiwan’s government is planning to permit permanent residency for mid-level skilled MWs, it will do so only under very strict requirements including a certain level of salary and professional certification, which makes the application process very challenging.

The rights and working conditions of low-skilled MWs in Korea and Taiwan have improved but at a slow pace and have still lagged, compared to high-skilled MWs in recent years (see Table 3). Korea permits MWs to shift to a new employer or new job three times of their own accord, but this free choice is not available to low-skilled MWs in Taiwan. Low-skilled MWs in Taiwan can shift to new jobs only under certain conditions caused by the employer, including performing illegal work outside the bounds of the permit and license, various kinds of abuse, and labor disputes. Without the free choice to transfer to a new job or employer, low-skilled MWs in Taiwan are regularly forced to call the 1955 telephone hotline and go through the labor dispute procedure. They may also seek assistance from MWs’ shelters, non-profit organizations, or local labor bureaus which exhaust both the official resources and MWs’ waiting time. Studies have shown that a free choice of employment for migrant care workers helps to stabilize MWs’ working lives and increases the stability of the employer hiring processes. Otherwise, without the freedom of choosing their employer/work, MWs may be trapped in forced labor and face potential exploitation; some MWs may even run away because employers have power over their rights to remain and work in the country [29]. Unlike Taiwan, Korea is a member of the UN and one of 13 nations to adopt 144 agreements that allow more flexibility when it comes to changing employment.

Another similarity between the two countries is that low-skilled migrant workers have organized labor unions, although these unions represent the rights for low-skilled workers, not high-skilled ones. MWs are allowed the right to organize labor unions by the Ministry of Labor in Taiwan, whereas it was strictly prohibited by the Ministry of Employment and Labor in Korea until the High Court allowed such labor unions in 2015. The two most vulnerable categories of MWs, including fishermen and live-in migrant care workers, established the Yilan Migrant Fishermen Union (YMFU) in 2013 and Domestic Caretakers Union Taoyuan (DCUT) in 2017, respectively, which are local grassroots unions organized by MWs with the local Taiwanese’s help. In the case of Korea, the Migrants Trade Union (MTC) was established in 2005, although it took around 10 years for it to become legal. The difference between the two countries is that low-skilled migrant workers performing different types of work have different visa categories and work permit periods. Labor unions in Korea are more likely to represent the comprehensive needs and demands of the current system. Interestingly, high-skilled workers have not organized a labor union in both Korea and Taiwan, mainly because the workers have better working conditions, freedom of choice, and employment protection. Their lives are also less likely to be restrained by restrictive migration policies.

Furthermore, low-skilled MWs are forced to leave their families and children back home because they cannot enjoy the right to family reunification in both Korea and Taiwan. The states also regulate low-skilled MWs’ visa status, working conditions, and length of stay. As for the right to vote for foreign residents including MWs, only foreigners in Korea are permitted to take part in local elections after receiving permanent residency. What needs to be focused on here is that this does not necessarily mean that all foreigners in Korea do have the freedom to pursue political participation. For example, according to the Access to Immigration Act in Korea and Immigration Act in Taiwan, foreigners are still not permitted to become involved in political activity [49]. That is why overseas students in Korea who are funded by the Korean government are required to sign a pledge that they will not participate in political activities such as establishing or joining political parties, participating in political demonstrations, publishing political articles, or making political declarations [46]. Being aware of such a prohibitive environment may lead migrants in all types of visa categories to feel inhibited about expressing their political opinions and voicing interest in changing discriminatory policies or laws.

We found a noticeable difference in the social protections for low-skilled MWs. Korea has unique insurance programs such as the Employment Permit System (EPS) designed for low-skilled MWs to make sure that MWs can claim their entire payment when they return to their home country after finishing their contract or changing their workplace after one year of work in Korea. However, in Taiwan, low-skilled MWs who have not received payments need to go through a labor dispute negotiation or even through legal disputes which may take a long time, and sometimes there is also a lack of evidence. In addition, under the principle of reciprocity, MWs from certain countries are entitled to national pension benefits in Korea; however, Taiwan has not signed bilateral agreements with its national pension plan. MWs in Taiwan, however, can reclaim their Labor Insurance contribution fee within five years of finishing their contract.

## 5. Conclusions

Overall, importing MWs is a strategy for both Korea and Taiwan to stimulate and maintain economic growth and industry competitiveness. Immigration regulation and alien control are still core elements of government policies to protect national security and social stability [50,51,52,53,54]. The governments of Korea and Taiwan are more willing to give civil and social rights to MWs, such as the freedom of speech and rights to file petitions, lawful assembly, and so on, but are less willing to grant rights to vote and participate in politics until foreigners obtain permanent residency. Adopting permanent residency to differentiate the “we-group” from the “they-group”, high-skilled MWs are more likely to be recruited and treated as the “we-group” compared to low-skilled MWs, who are considered as the supplemental labor force [51,52,53,54]. However, the government of Korea and Taiwan are gradually moving in the direction of immigration integration and protection under the pressure of social movements from inside and outside the countries. Both governments emphasize immigration control/regulation over integration since national security and the protection of citizens are the priorities; however, strict regulations on low-skilled MWs have resulted in a growing number of undocumented MWs in Korea and an increasing number of disconnected/run-away MWs in Taiwan, which challenge both nations’ management of MWs. For those low-skilled MWs, maintaining legal status is a significantly important way that both the Korean and Taiwanese governments are willing to meet their basic needs, working conditions and protections. At once becoming illegal/runaway/undocumented, those unaccounted-for MWs are treated as non-economic immigration and will face deportation and restricted re-entry. However, for those live-in migrant care workers, running away seems like a soundless scream and it is a strategy showing the only way of mobility (to be better heard, with a better salary and wellbeing) from the immobility embedded in policy regulations and structural obstacles such as being stuck within the precarity of labor, including long working hours, vulnerable working conditions, low salaries, limited holidays, no privacy in living environments, and without protections within the law. For Korea and Taiwan, under the trend of globalization, attracting high-skilled and low-skilled MWs will become more challenging due to competition with exporting countries, boycotts of the sending countries, and the advocation of MWs’ rights from local and international labor unions and the UN.

## Figures and Tables

**Figure 1 ijerph-19-16270-f001:**
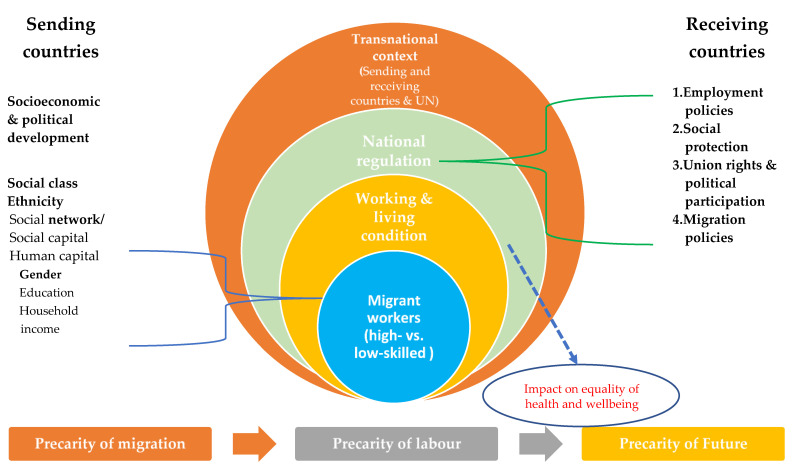
Research conceptual model.

**Table 1 ijerph-19-16270-t001:** Foreigner workers in Korea and Taiwan (Unit: person).

		Foreigner Workers in Korea			Foreigner Workers in Taiwan		
	White Collar/Professional Workers ^1^	Blue Collar/Low Skilled Workers ^2^			White Collar/Professional Workers ^4^	Blue Collar/Low Skilled Workers	
		Industrial (E-9)	Ethnic Koreans (H-2)	Crew Employees (E-10)		Industrial Sector ^5^	Welfare Sector ^6^
2002	15,869	128,229 ^3^	-	-	25,933	182,937	120,711
2006	29,011	161,867	29,574	-	29,336	184,970	153,783
2010	43,608	220,319	282,662	6716	26,589	193,545	186,108
2017	47,404	279,127	238,880	16,069	30,927	425,985	250,157
2019	47,000	276,755	226,322	17,603	31,125	456,601	261,457
2020	43,000	236,950	154,537	17,552	39,522	457,267	251,856
2021	45,000	217,729	125,493	17,921	45,300	443,104	226,888

Source: Taiwan’s statistics are from the Ministry of Labor; South Korea’s statistics are from the Ministry of Justice. Note: ^1^ In Korea, professional visas include the E-1 (professor visa), E-2 (long-term visa to teach a foreign language), E-3 (research), E-4 (technological guidance), E-5 (specialty occupation), E-6 (culture and entertainment), and E-7 (particular occupation) visas. ^2^ Non-professional visas include the E-9 (EPS workers) and E-10 (crew employee) visas. ^3^ The number includes regular and irregular migrant workers. ^4^ The number of white collar/professional workers in Taiwan refers to traders, engineers, teachers, missionaries, and technicians; this category includes specialized or technical workers, full-time teachers teaching courses in foreign languages in short-term classes registered for supplementary schooling, school teachers, directors of businesses invested in or set up by overseas Chinese or foreigner(s), religious, artistic, and show business workers, and sports coaches and athletes. ^5^ Industrial workers refer to low-skilled workers in the industrial sector. ^6^ Welfare workers refers to care workers in the welfare sector, and typically refers to live-in migrant care workers in private households or in nursing homes and institutions.

**Table 2 ijerph-19-16270-t002:** Extension of stays for low-skilled MWs in Taiwan.

Year	Basic Stay	Extension	Total Stay (Years)	Required Departure
**8 May 1992**	2 years	1 Year	3	NA
**21** **May 1997**	3 years	2 Years	5	NA
**21 January 200**	3 years	3 Years	6	40 days
**13 May 2003**	3 years	3 Years	6	1 day
**11 July 2007**	3 years	3 Years (2 times)	9	1 day
**30 January 2012**	3 years	3 Years (3 times)	12	1 day
**9 October 2015**	3 years	3 Years (4 times)	14 ^1^	1 day ^2^

Source: The Employment Service Act [49]. Note: ^1^ Only migrant care workers qualify for 14 year stays; industrial production workers can stay for a maximum of 12 years. ^2^ On 2 November 2016, the one-day departure rule was cancelled.

**Table 3 ijerph-19-16270-t003:** Comparison of employment policies, union rights, political participation, social protection and migration policies for low-skilled MWs in Korea and Taiwan.

Countries	Korea	Taiwan	
Different Policy		Industrial Sectors	Welfare Sectors
** *Employment* **			
Working conditions Working hours	Minimum standard (factory) 11.74 h	Minimum standard (factory) 9.76 h	Invisible (private household) 10.2 h
Salary	USD 1555 in 2016	USD 770 in 2019 (minimum wage)	USD 567 in 2019 (under minimum wage)
Training and employment services	Factory-based	Factory-based	Limited
Shift to new job	Yes (three times)	No (exception)	
** *Union rights* **	No to Yes (legalized until 2015)	Yes (2013 YMFU; 2017 DCUT)	
** *Political participation* **	Yes	No	
** *Social protection* **			
Employment insurance	Yes	Yes	No
Industrial accident Compensation insurance	Yes	Yes	No
Pension	National pension	Labor insurance	No
Health rights	National health insurance	National health insurance	
** *Permanent residency* **	Very limited	No	No
Family reunification	No	No	No

## Data Availability

Not applicable.

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
