# Peer review of "Exclusion or Inclusion: National Differential Regulations of Migrant Workers’ Employment, Social Protection, and Migrations Policies on Im/Mobilities in East Asia-Examples of South Korea and Taiwan"

_ijerph, 2022, doi:10.3390/ijerph192316270_

Round 1

Reviewer 1 Report

I found this article to be informative and will be beneficial to scholars studying migration policy. It was well-structured. I have only minor comments and I will list them below:

- Is there a research question? If so, then the authors should change the Section Title "Results"

- I would like to see a summary of migration policy in Taiwan and South Korea before the details of the policies. Are we seeing a convergence in demand for high- and low-skill migrant workers? Meanwhile, is there a divergence in policy between low-skill and high-skill migrant workers? 

- The table on p.12 needs to be clarified. What country is being referred to? While industrial and welfare sectors make up the column, the paper did not emphasize the sectors on focused more on the type of labor. 

I think this paper meets the goal of describing in detail the comparisons and differences in migration policy between low-skill and high-skill workers in East Asia. 

Author Response

Response to Reviewer 1 Comments

Point 1: I found this article to be informative and will be beneficial to scholars studying migration policy. It was well-structured. I have only minor comments and I will list them below:

- Is there a research question? If so, then the authors should change the Section Title "Results".

- I would like to see a summary of migration policy in Taiwan and South Korea before the details of the policies. Are we seeing a convergence in demand for high- and low-skill migrant workers? Meanwhile, is there a divergence in policy between low-skill and high-skill migrant workers? 

- The table on p.12 needs to be clarified. What country is being referred to? While industrial and welfare sectors make up the column, the paper did not emphasize the sectors on focused more on the type of labor. 

I think this paper meets the goal of describing in detail the comparisons and differences in migration policy between low-skill and high-skill workers in East Asia. 

Response 1: We really appreciate the reviewer’s delightful recommendations which improve the quality of our article.

-We have tried to demostrate our research question clearly as line 55-68 and showed our research results in the title as “Exclusion for low-skilled migrant workers and inclusion for high-skilled workers”.

-We have added a brief summary of migration policy in South Korea and Taiwan as line 127-154 showed. We can see a convergence of recruiting large amount of migrant workers due to the shortage of labour regardless of high-skilled and low-skilled migrant workers as well as the “class selection” and a global “convergnece” of immigration policies in line 155-159. We also see a difference in social protection between Korea and Taiwan as line 700-709 discussed. The big divergence is Taiwan government starts to consdier gradually lower the threshold of salary to open permanent residence to median blue-colar migrant workers with certificate and plenty years of experience as line 549-557 discussed.

-We had clarified Talbe 2 refer to MWs in in Taiwan in the title as line 545 showed. We also explain in the note1 “Only migrant care workers qualify for 14-year stays; industrial production workers can stay for a maximum of 12 years” as line 550-551 showd.

Reviewer 2 Report

The subject of the article is interesting, and it is linked to the objectives of the journal, however, there are some issues that have to be reconsidered.

For better visibility on databases, the authors are asked not to repeat the words/concepts included in the article's title among keywords.

I suggest that the part called Introduction should be enriched with studies that reflect the Literature Review in the field 

The Methodology part is relatively superficial and has to be reconsidered to make the study replicable.

The article is difficult to follow and its structure is not clear.

Table 3 is part of the Results, not of the Discussions.

I suggest fully reconsidering the article.

Author Response

Response to Reviewer 2 Comments

Point 1: The subject of the article is interesting, and it is linked to the objectives of the journal, however, there are some issues that have to be reconsidered.

For better visibility on databases, the authors are asked not to repeat the words/concepts included in the article's title among keywords.

Point 2: I suggest that the part called Introduction should be enriched with studies that reflect the Literature Review in the field. 

Point 3: The Methodology part is relatively superficial and has to be reconsidered to make the study replicable.

The article is difficult to follow and its structure is not clear.

Table 3 is part of the Results, not of the Discussions.

I suggest fully reconsidering the article.

Response 1: We really appreciate the reviewer’s wonderful suggestions and we have deleted the keywords “migrant workers”, “migration policies”, and “regulation” to avoid the repetition of the words/concepts in the title.

Response 2: We have also added a few cruical articles, including Messey et al. (1993), Brettell and Hollifield Eds. (2015), and Schewel (2020) to sythesize migration theory and tried to enrich literature review context (please see line 34-42 and 46-54, and also see line 712-714 and 722-724) . 

Response 3: We tried to draw our research conceptual model as line 55-68 and Figure 1 showed in line 98 to clarify how we frame our article and reinforce the thinking context of our research methods (in line 104-109 ) and explain how we formulate our comparative perspective as well as hope to make the structure of our arguments clearer. We really appreciate reviewer’s reminder that Table 3 is a summary from our results, which we hope to facilitate our discussions part. It is also why we summarize our comparative findings in Table 3 to make future potential comparative analysis practicable and could be duplicated. We sincerely hope that the reviewer can see our efforts on revision to improve the quality of our article.

Reviewer 3 Report

This is a very nicely written paper. I would like to suggest the authors the following to further improve the paper.

i) Elaborate the method section. Explain how the information were collected, synthesized, and analyzed.

ii) Readers would be interested to know (however, it appears to be beyond the scope of this paper) whether and how the labor policies of these two countries (e.g., legal status and other services such as working and employment conditions, social protection, union rights and political participation, and access to permanent residency) have affected their working efficiency and their own life.

Overall, it is a great work. 

Author Response

Response to Reviewer 3 Comments

Point 1: This is a very nicely written paper. I would like to suggest the authors the following to further improve the paper.

  1. i) Elaborate the method section. Explain how the information were collected, synthesized, and analyzed.
  2. ii) Readers would be interested to know (however, it appears to be beyond the scope of this paper) whether and how the labor policies of these two countries (e.g., legal status and other services such as working and employment conditions, social protection, union rights and political participation, and access to permanent residency) have affected their working efficiency and their own life.

Overall, it is a great work. 

Response 1: Thanks for the reviewer’s encouragement and compliment.

  1. i) We have tried to enrich our literature part in our instroduction (please see line 34-42 and 46-54, and also see line 712-714 and 722-724) to sythesize migration theory and tried to draw our research conceptual model (as Figure 1 showed in line 98) to reinforce and deliberate our methodolgoy part (in line 104-109 ) and explain how we collect information, analyize data, and evaluate comparative studies. We hope to make the structure of our article and comparative context clearer.
  2. ii) We aslo added how structual obstacles and policy regulations affect MWs’ own life in the conclusion part as line 700-709 showed. We look forward to initiate an article to elaborate the how structural obstackes have tremendously impacts on the precarity of MWs’ current and future life as our next goal.

Round 2

Reviewer 2 Report

The authors succeeded in answering my concerns, the manuscript can be published.